# GENERALIZED HEAVY-TAILED MUTATION FOR EVOLUTIONARY ALGORITHMS

**Anton V. Eremeev, Dmitri V. Silaev & Valentin A. Topchii**

Novosibirsk State University
1, Pirogova str., Novosibirsk, 630090, Russia,
Sobolev Institute of Mathematics
4, Koptyuga pr., Novosibirsk, 630090, Russia
eremeev@ofim.oscsbras.ru

## ABSTRACT

The heavy-tailed mutation operator, proposed by Doerr, Le, Makhmara, and Nguyen (2017) for evolutionary algorithms, is based on the power-law assumption of mutation rate distribution. Here we generalize the power-law assumption using a regularly varying constraint on the distribution function of mutation rate. In this setting, we generalize the upper bounds on the expected optimization time of the $(1 + (\lambda, \lambda))$ genetic algorithm obtained by Antipov, Buzdalov and Doerr (2022) for the OneMax function class parametrized by the problem dimension $n$. In particular, it is shown that, on this function class, the sufficient conditions of Antipov, Buzdalov and Doerr (2022) on the heavy-tailed mutation, ensuring the $O(n)$ optimization time in expectation, may be generalized as well. This optimization time is known to be asymptotically smaller than what can be achieved by the $(1 + (\lambda, \lambda))$ genetic algorithm with any static mutation rate. A new version of the heavy-tailed mutation operator is proposed, satisfying the generalized conditions, and promising results of computational experiments are presented.

## 1 INTRODUCTION

A distinctive feature of evolutionary algorithms (EA) is their imitation of the process of evolutionary adaptation of a biological population to environmental conditions. Individuals correspond to trial points in the solution space of an optimization problem, and their fitness is determined by the values of the objective function, taking into account penalties for violating the problem's constraints, if any. The construction of new trial points in EA is accomplished using mutation and crossover operators. When using the latter, EAs are commonly referred to as genetic algorithms. When solving unconstrained pseudo-Boolean maximization problems $\max\{f(x) : x \in \{0,1\}^n\}$, or minimization problems $\min\{f(x) : x \in \{0,1\}^n\}$, where the objective function is $f : \{0,1\}^n \to \mathbb{R}$, one of the most frequently used mutation operators is the *standard mutation* Goldberg (1989), where each bit of the given string $x \in \{0,1\}^n$ independently changes its value with a given probability $p$. In this paper, we will assume that in the case of standard mutation, at each iteration of an EA with distribution $\mathrm{Bin}(n, p)$, a number of mutated bits $\ell$ is selected, and the next descendant is obtained from the parent solution by making changes to $\ell$ randomly selected bits.

The main performance characteristic of evolutionary algorithms in solving optimization problems is the *optimization time*, hereafter denoted by $T$, which is defined as the number of times the fitness function is evaluated until the optimum is reached for the first time. Typically, the expected value of the optimization time or the average optimization time is studied. In this paper, we study the optimization time of a genetic algorithm with a computational scheme from Antipov et al. (2022) when maximizing the fitness function $\mathrm{ONEMAX}(x) = \sum_{i=1}^{n} x_i$. A family of such functions, parameterized by the problem dimension $n$, is one of the basic benchmarks in the theory of evolutionary computation, used to evaluate the effectiveness of EAs on simple problems. In particular, in the case of high expected optimization time on $\mathrm{ONEMAX}$, random search algorithms or subsets of their adjustable hyper-parameter values are considered ineffective Lehre (2011); Oliveto & Witt (2015).

Symmetry considerations imply that EAs based on the standard mutation behave identically on both ONEMAX and any function $\text{ONEMAX}_z : \{0,1\}^n \to \mathbb{R}$, defined as

$$\text{ONEMAX}_z(x) := |\{i \in \{1, ..., n\} : x_i = z_i\}|$$

for any bit string $z \in \{0,1\}^n$. It was shown by Erdős & Rényi (1963) that any algorithm that obtains information only from queries of the fitness function $\text{ONEMAX}_z$ will, on average, require $\Omega(n/\log n)$ fitness function computations before first hitting the optimal solution $z$, and this bound cannot be improved.

Doerr et al. (2015) developed a genetic algorithm, $(1 + (\lambda, \lambda))$ GA, with a new crossover operator that eliminates "bad" mutations. At each iteration of the algorithm, $(1 + (\lambda, \lambda))$ GA, a single parent individual generates $\lambda = \lambda(n)$ offspring independently of each other at equal Hamming distance $\ell$ from the parent, with $\ell \sim \text{Bin}(n, p)$. Next, the best-fitting of these solutions is selected, and a crossover operator with parameter $c$ is applied to it. With probability $c$, the crossover operator uses bits from the best offspring, and with probability $1 - c$, it uses bits from the parent solution. In this way, $\lambda$ individuals are created, and the best of these $\lambda$ individuals is accepted as a new parent if it is at least as fit as the previous parent. A theoretical analysis of the optimization time has shown that the $(1 + (\lambda, \lambda))$ GA algorithm, for many values of the adjustable parameters, is asymptotically faster by ONEMAX than most classical evolutionary algorithms.

As demonstrated by Doerr et al. (2017), choosing a mutation parameter for multi-extremal problems is significantly more challenging than for ONEMAX. To overcome this difficulty, Doerr et al. (2017) proposed using a random choice for the mutation parameter $p$ following a heavy-tailed distribution, namely, a truncated power-law distribution with exponent $-\beta < -1$. In this case, a random number $\alpha \in \{1, \ldots, \lfloor \frac{n}{2} \rfloor\}$ is first selected, so that the probability of selecting $\alpha = k$ is proportional to $k^{-\beta}$, $k \leq \lfloor \frac{n}{2} \rfloor$. Then, $p = \alpha/n$ is set and the standard mutation is applied with this value $p$. At each iteration of the EA, the values of $\alpha$ are selected independently.

As shown by Doerr & Doerr (2018) for the ONEMAX case, the optimal choice of a fixed value of the mutation parameter p for the entire running time of the GA yields an optimization time of

$$E[T] = \Theta\left(n\sqrt{\frac{\log(n)\log\log\log(n)}{\log\log n}}\right), \tag{1}$$

which is asymptotically smaller than the average optimization time of many known evolutionary algorithms.

Antipov et al. (2022) demonstrated the effectiveness of fast mutation in optimizing the ONEMAX function. Here, the $(1 + (\lambda, \lambda))$ GA from Doerr et al. (2015), combined with the fast mutation operator, was considered. Antipov et al. (2022) proved an upper bound for the average optimization time of order $O(n)$ for $(1 + (\lambda, \lambda))$ GA with fast mutation on ONEMAX, under a specific choice of the distributions of the random variables $\lambda$ and $p$. This is less than the average optimization time of $(1 + (\lambda, \lambda))$ GA with any fixed probability of mutations. In this algorithm, both the population size $\lambda$ and the rapid mutation parameter $p$ have a truncated power-law distribution with upper bounds $\lambda \leq u_n$ and $p \leq u_n/n$, respectively. The linear bound of Antipov et al. (2022) holds when the power-law exponent $\beta$ satisfies the inequalities $2 < \beta < 3$ and $u_n \geq \ln^{1/(3-\beta)} n$.

**The main result** of this work, presented in Theorems 1 and 2, shows that the upper bounds on the expected optimization time of $(1 + (\lambda, \lambda))$ GA, similar to those obtained in (Antipov et al., 2022), hold not only for truncated power-law distributions of the random variables $\lambda$ and $p$, but also for a wider class of distributions described in terms of regularly varying constraints on the distribution function of this quantity. As follows from (1), the linear bound on the average optimization time obtained by us, like the linear bound from (Antipov et al., 2022), turns out to be asymptotically smaller than the average optimization time of $(1 + (\lambda, \lambda))$ GA with any mutation parameter $p$ that remains unchanged during the algorithm's execution.

The proofs are provided in the Appendix.

A special case of this result, obtained without using the apparatus of regularly varying functions, is presented in (Eremeev & Topchii, 2024).

## 1.1 Notation and definitions

Denote $\mathbb{N}_m := \{k : k \in \mathbb{N}, k \leq m\}$, $S := \{0,1\}^n$, $|S| = 2^n$. Introduce the norm and Hamming distance $|x| = \sum_{i=1}^n x_i$ and $|x - y| = \sum_{i=1}^n |x_i - y_i|$ for $x, y \in S$. Denote by $x^*$ the solution with ones in all positions, and $Z_s = \{x \in S : |x - x^*| = s\}$ is the set of solutions having exactly $s$ zeros, $s = 0, \ldots, n$. In particular, $Z_0 = \{x^*\}$.

Let $\lambda(n)$, $n \in \mathbb{N}$ be a set of jointly independent random variables (r.v.s) with ranges from a subset of $\mathbb{N}_{u_n}$, where $u_n \leq 0.5n$ is the maximum possible value of $\lambda(n)$ with positive probability. In other words, probabilities $\mathbf{P}(\lambda(n) = k) = p_{n,k} \geq 0$ only for $k \in \mathbb{N}_{u_n}$ and $p_{n,u_n} > 0$. Constraints on $p_{n,k}$ will be introduced later. The r.v.s $\lambda_*(n)$ with different indices instead of $*$ are independent and identically distributed with $\lambda(n)$. We study the case $u = u(n) \to \infty$ as $n \to \infty$.

Let us describe the algorithm $\mathcal{A}$ under study, whose predecessor is the $(1 + (\lambda, \lambda))$ EA from (Doerr et al., 2015) with deterministic mutation probabilities $p = \lambda(n)/n$ and crossover parameter $c = \lambda^{-1}(n)$. Later in (Antipov et al., 2022), a randomization of the $(1 + (\lambda, \lambda))$ EA from (Doerr et al., 2015) was proposed with respect to the population size $\lambda(n)$, where the probabilities $\mathbf{P}(\lambda(n) = k) = p_{n,k}$ are expressed via power functions. This algorithm in (Antipov et al., 2022) was named the *fast* $(1 + (\lambda, \lambda))$ *genetic algorithm*. When the parameter $\lambda(n)$ is chosen according to a power law and the mutation parameter $p = \lambda(n)/n$, the algorithm from (Antipov et al., 2022) has a heavy-tailed mutation, coinciding with that proposed in (Doerr et al., 2017). In this paper, we abandon explicit expressions for $p_{n,k}$ and provide only constraints on the distribution function of the r.v. $\lambda(n)$. In all other respects, algorithm $\mathcal{A}$ coincides with the fast $(1 + (\lambda, \lambda))$ genetic algorithm.

*Algorithm $\mathcal{A}$: $(1 + (\lambda, \lambda))$ genetic algorithm with regularly varying constraints on the distribution function of $\lambda(n)$ with an upper limit of its values $u_n$, maximizing the function $f : \{0,1\}^n \to \mathbb{R}$*

1. $x \leftarrow$ random bit string of length $n$;

2. **while** *not terminated* **do**

3.     Choose $\lambda$ from [1..u] with $\mathbf{P}(\lambda = k) = p_{n,k}$;

4.     Choose $\ell \sim \text{Bin}(n, \lambda(n)/n)$;

5.     **for** $i \in [1..\lambda]$ **do**

6.         $x^{(i)} \leftarrow$ a copy of $x$;

7.         Flip $\ell$ bits in $x^{(i)}$ chosen uniformly at random;

8.     **end**

9.     $x' \leftarrow \arg\max_{z \in \{x^{(1)}, \ldots, x^{(\lambda)}\}} f(z)$;

10.     **for** $i \in [1..\lambda]$ **do**

11.         Create $y^{(i)}$ by taking each bit from $x'$ with probability
$\lambda^{-1}$ and from $x$ with probability $(\lambda - 1)\lambda^{-1}$;

12.     **end**

13.     $y \leftarrow \arg\max_{z \in \{y^{(1)}, \ldots, y^{(\lambda)}\}} f(z)$;

14.     **if** $f(y) \geq f(x)$ **then**

15.         $x \leftarrow y$;

16.     **end**

17. **end**

Algorithm $\mathcal{A}$ starts with a random initial bit string $x$. Each iteration of the algorithm consists of obtaining a random realization of the r.v. $\lambda(n)$ for the population size $\lambda$, followed by a mutation phase, a crossover phase, and a selection phase. In the mutation phase (lines 4–8), after obtaining a realization $\ell$ with distribution $\text{Bin}(n, p)$, $\lambda$ offspring are created from $x$ by making changes in $\ell$

randomly selected bits in each of them. From these new $\lambda$ individuals, one with the highest fitness is selected for further participation in the crossover phase. If there is more than one offspring with maximum fitness, we choose one of them uniformly at random. Denote the selected offspring by $x'$. In the crossover phase (lines 10–12), $\lambda$ new offspring are created from the parent individual $x$ and the winner $x'$ from the mutation phase. From these $\lambda$ offspring, the string with the highest fitness is selected. If there are several, then we choose uniformly at random one of them (strings that coincide with the parent solution $x$ are not considered). In the selection phase (line 14), the parent $x$ is replaced by the winner individual of the crossover phase $y$, if the fitness of $y$ is not lower than the fitness of $x$. In algorithm $\mathcal{A}$, as in many other algorithms mentioned above, no stopping criterion is specified. This is because in theoretical studies we are mainly interested in the time of first reaching the optimum. In practical applications of algorithm $\mathcal{A}$, it is natural to specify a stopping criterion.

## 2 MAIN PROPERTIES OF THE ALGORITHM

Denote by $\ell_{\lambda(n)}(s)$ the number of iterations of Algorithm 1 from (Doerr et al., 2015), starting from a solution $x \in Z_s$, until the first improvement of the fitness function for fixed $n$ and $\lambda(n)$, and the probability of improvement on the first iteration by $p_{\lambda(n)}(s) = \mathbf{P}(\ell_{\lambda(n)}(s) = 1)$. Further we study algorithm $\mathcal{A}$ with the fitness function ONEMAX, and the probabilities $p_{\lambda(n)}(s)$ coincide for all $x \in Z_s$. Note that with a random choice of $\lambda(n)$, the r.v. $\ell_{\lambda(n)}(s)$ does not carry a semantic load, but only the event $\{\ell_{\lambda(n)}(s) = 1\}$ is important.

We present Lemma 7 from (Doerr et al., 2015) with a fixed (not random) value of $\lambda$ for each $n$ with mutation parameter $p = \lambda/n$ and crossover parameter $c = \lambda^{-1}$.

**Lemma 1** *One iteration of algorithm $\mathcal{A}$ with fixed $\lambda$ in the case of the fitness function ONEMAX, starting from a solution $x \in Z_s$, leads to an improvement of the current fitness function value with probability satisfying the inequality*

$$p_\lambda(s) \geq C \left( 1 - (1 - s/n)^{\lambda^2/2} \right) \left( 1 - e^{-1/8} \right) \tag{2}$$

*for some constant $C > 0$ independent of $n$.*

Further, without additional comments, for constants independent of $n$ we will use the notations $C > 0$ and $c > 0$ with or without indices. These constants may be interrelated, but only the presence of these positive constants is fundamentally important. In some cases, to simplify the presentation, for different constants we will use the symbols $C^*$ and $c^*$ without indicating their explicit form, and even in one equation, in different parts of the equality, these values may be different.

Inequality (2) taking into account the bound

$$1 - (1 - p)^\lambda \geq \frac{\lambda p}{1 + \lambda p}, \ \forall p \in (0, 1), \ \lambda > 0,$$

from Lemma 2 of (Antipov et al., 2022) can be written as

$$p_{\lambda(n)}(s) \geq C_1 \frac{0.5\lambda^2(n)s/n}{1 + 0.5\lambda^2(n)s/n}, \tag{3}$$

in particular, inequality (3) can be written as

$$p_{\lambda(n)}(s) \geq C_2 \lambda^2(n)s/n, \text{ for } \lambda^2(n)s/n < 1, \tag{4}$$

$$p_{\lambda(n)}(s) \geq C_3, \text{ for } \lambda^2(n)s/n \geq 1. \tag{5}$$

Let $\lambda(n)$ be random. Formally, at each iteration with number $t$, a r.v. $\lambda_t(n)$ is realized with the same distribution as $\lambda(n)$. Given that there is no dependence on $t$, we omit this index. Using the law of total probability (averaging over $\lambda(n)$), denote the probability of improvement in one iteration by $p_n(s) = \mathbf{E}_{\lambda(n)} p_{\lambda(n)}(s) = \mathbf{P}(\ell_n(s) = 1)$, where $\ell_n(s)$ is the number of iterations until improvement of the fitness function. Formally, in the definition of the probability $p_n(s)$, which does not depend on the iteration number $t$, the event $\{\ell_n(s) = 1\}$ depends on this number $t$, and $\ell_n(s)$ equals the number of unsuccessful iterations in a series of iterations until the first successful one. The count of iterations starts from the initial individual or after the next increase in the fitness function.

Compute the expectation $\mathbf{E}\ell_n(s)$. The random variable $\ell_n(s)$ has a geometric distribution with parameter $p_n(s)$. By definition we have

$$
\begin{aligned}
\mathbf{P}(\ell_n(s) = k) &= (1 - p_n(s))^{k-1} p_n(s), \ k \in \mathbb{N}, \\
\mathbf{E}\ell_n(s) &= (1 - p_n(s)) p_n^{-1}(s) + 1 = p_n^{-1}(s).
\end{aligned}
\tag{6}
$$

Suppose that for some fixed $m_0, m_1 \in \mathbb{N}$ there exists a constant $c > 0$ independent of $n$ such that for all $n \in \mathbb{N} \backslash \mathbb{N}_{m_1}$ the inequalities hold

$$
\mathbf{P}(\lambda(n) \in \mathbb{N}_{m_0}) \geq c.
\tag{7}
$$

A sufficient condition for inequality (7) to hold is uniform in $n$ boundedness of $\mathbf{E}\lambda(n)$, i.e., the conditions $\mathbf{E}\lambda(n) \leq c_1, \ \forall n \in \mathbb{N}$, for some $0 < c_1 < \infty$. Then by Markov's inequality for non-negative random variables $\lambda(n)$ for any $c_0 > c_1$

$$
\mathbf{P}(\lambda(n) < c_0) \geq 1 - \mathbf{E}\lambda(n)/c_0 \geq 1 - c_1/c_0 > 0.
$$

Conditions for inequality (7) to hold will be formulated as: for sufficiently large $n$. Obviously, if inequality (7) holds for some fixed $m_0$ and $m_1$, then it also holds when these values are replaced by any fixed values greater than them.

Estimate the fraction on the right-hand side of bound (3) for $\lambda(n) \in \mathbb{N}_{m_0}$. If $s \leq 0.5 n m_0^{-2}$, then $0.5\lambda^2(n)s/n \leq 0.25$ and the bound holds

$$
\frac{0.5\lambda^2(n)s/n}{1 + 0.5\lambda^2(n)s/n} \geq 0.4\lambda^2(n)s/n \geq 0.4sn^{-1}.
$$

Otherwise, $s > 0.5 n m_0^{-2}$ and $(0.5\lambda^2(n)s/n)^{-1} < 4$ and the bound holds

$$
\frac{0.5\lambda^2(n)s/n}{1 + 0.5\lambda^2(n)s/n} > 0.2 = 0.2sn^{-1}s^{-1}n \geq 0.2sn^{-1}.
$$

When inequality (7) holds, due to the last two bounds and inequality (3), for all $n \in \mathbb{N} \backslash \mathbb{N}_{m_1}$ and $s \in \mathbb{N}_n$ the inequalities hold

$$
p_n(s) \geq \mathbf{E}_{\lambda(n)}\{p_{\lambda(n)}(s); \lambda(n) \in \mathbb{N}_{m_0}\} \geq \frac{0.2C_1 s}{n} \mathbf{P}(\lambda(n) \in \mathbb{N}_{m_0}) = \frac{C_4 s}{n}.
\tag{8}
$$

**Lemma 2** *In the case of the fitness function* ONEMAX, *the average number of iterations of algorithm* $\mathcal{A}$, *starting from a solution* $x \in Z_s$, *until improvement of the objective function, under condition (7) satisfies the inequality*

$$
\mathbf{E}\ell_n(s) \leq C_4^{-1} n/s, \ s \in \mathbb{N}_n,
\tag{9}
$$

*where $C_4$ is defined in expression (8).*

Let $\tau_i(n)$ be the number of iterations until the first hit of $x^*$ given that the process starts from an individual $x^{(0)} \in Z_i$. If the initial solution is chosen uniformly from the population, then

$$
\mathbf{P}\left(x^{(0)} \in Z_i\right) = C_n^i 2^{-n}, \quad \mathbf{E}\tau_i(n) \leq \sum_{s=1}^{i} \mathbf{E}\ell_n(s) = \sum_{s=0}^{i} \mathbf{E}\ell_n(s).
\tag{10}
$$

Let $\tau(n)$ be the number of iterations until the first hit of $x^*$ from a randomly chosen individual $x^{(0)}$. By the law of total probability, the representation holds

$$
\begin{aligned}
\mathbf{E}\tau(n) &= 2^{-n} \sum_{i=0}^{n} C_n^i \mathbf{E}\tau_i(n) \leq 2^{-n} \sum_{i=0}^{n} C_n^i \sum_{s=0}^{i} \mathbf{E}\ell_n(s) \\
&= 2^{-n} \sum_{s=1}^{n} \mathbf{E}\ell_n(s) \sum_{i=s}^{n} C_n^i = 2^{-n} \sum_{s=1}^{n\epsilon} \mathbf{E}\ell_n(s) \sum_{i=s}^{n} C_n^i
\end{aligned}
$$

$$+ \quad 2^{-n} \sum_{s=n\epsilon+1}^{n} \mathbf{E}\ell_n(s) \sum_{i=s}^{n} C_n^i \leq \sum_{s=1}^{n\epsilon} \mathbf{E}\ell_n(s) + C_5 n, \tag{11}$$

where $\epsilon \in (0, 1)$ is an arbitrary fixed number and the sum of combinations divided by $2^n$ is the sum of probabilities, which does not exceed 1, and $\mathbf{E}\ell_n(s) \leq C_4^{-1}\epsilon^{-1} =: C_5$ for $n\epsilon + 1 \leq s \leq n$. Here and below, in sums over subsets of natural numbers and functions of a natural argument with not necessarily integer limits or values, we consider that these quantities are equal to the nearest smaller integer.

## 3 AVERAGE NUMBER OF ITERATIONS UNTIL HITTING THE OPTIMUM

### 3.1 REGULARLY VARYING FUNCTIONS

Let us consider several definitions and properties of regularly varying functions. (See e.g. the monograph (Seneta, 1976) and §9 of Chapter VIII in (Feller, 1971).)

**Definition 1** *A measurable function $g(v) > 0$, defined for sufficiently large $v \in \mathbb{R}^+$ or $v \in \mathbb{N}$, is called **regularly varying at infinity** with exponent $\alpha \in \mathbb{R}$, if for any fixed $c \in \mathbb{R}^+$ the condition holds*

$$\lim_{v \to +\infty} \frac{g(cv)}{g(v)} \to c^\alpha, \tag{12}$$

*where $cv$ should be replaced by $[cv]$, the integer part of the number in the case $v \in \mathbb{N}$.*

*In the case $\alpha = 0$, the function is called **slowly varying at infinity**.*

The property of regular variation is asymptotic and the function $g(v) > 0$ does not necessarily have to be defined on any initial interval of the half-axis. Regularly varying functions for $v \in \mathbb{N}$ and $v \in \mathbb{N}$ will be called regularly varying sequences. (There is another definition of regularly varying sequences for $c \in \mathbb{N}$ in expression (12), but to avoid unnecessary complications, we restrict ourselves to the simple case.)

Any regularly varying at infinity function $g(v)$, $v \in \mathbb{R}^+$, will be asymptotically equivalent to a regularly varying at infinity step function $g([v])$, which can be interpreted as a regularly varying at infinity sequence $g(z)$, $z \in \mathbb{N}$. Obviously, the converse statement also holds.

**Lemma 3** *For a regularly varying at infinity sequence $g(z)$, $z \in \mathbb{N}$, there exists a regularly varying at infinity function $\hat{g}(v)$, $v \in \mathbb{R}^+$, such that $g([v]) = \hat{g}([v])$ for sufficiently large $v \in \mathbb{R}^+$.*

Further, we will not distinguish between notations for sequences and functions defined through each other, i.e., we identify the symbols $\hat{g}$ and $g$.

**Definition 2** *A measurable function $g(v) > 0$, defined for sufficiently small $v \in \mathbb{R}^+$, is called **regularly varying at zero** from the right with exponent $\alpha \in \mathbb{R}$, if for any fixed $c \in \mathbb{R}^+$ the condition holds*

$$\lim_{v \to +0} \frac{g(cv)}{g(v)} \to c^\alpha$$

*In the case $\alpha = 0$, the function is called **slowly varying at zero** from the right.*

Definition (2) can be formulated for any $a \in \mathbb{R}$, both in one-sided and two-sided variants, by replacing $g(v) > 0$ with $g(v - a) > 0$ and convergence in one sense or another of $v - a$ to zero.

Here are several examples of regularly varying functions. This class is a generalization of the family of power functions $g(v) = v^\alpha$, $v \in \mathbb{R}^+$, and sequences for $v \in \mathbb{N}$. Functions $v^\alpha$, $v \in \mathbb{R}^+$, for any $\alpha \in \mathbb{R}$ are regularly varying at 0 and at infinity. Functions $|\ln v|$, $|\ln|\ln v||$ and any powers of them are slowly varying at 0 and at infinity. If a slowly varying function at 0 or at infinity is denoted by $\ell(v)$, then for any $\alpha \in \mathbb{R}$ the function $g(v) = v^\alpha \ell(v)$ will be regularly varying at zero or at infinity with exponent $\alpha$. Moreover, all regularly varying functions are representable only in this form.

### 3.2 GENERALIZATION OF CONDITIONS ON POPULATION SIZE AND MUTATION SETTINGS

The following two definitions specify our conditions on the distribution of the population size $\lambda(n)$ (and hence on the distribution of the mutation parameter $p = \lambda(n)/n$).

**Definition 3** *A random sequence $\lambda(n)$ satisfies condition $\mathcal{A}_2^c$ if, for all $n \in \mathbb{N}$, the second moment satisfies the inequality $\mathbf{E}\lambda^2(n) < C$ for some constant $C > 0$.*

**Definition 4** *A random sequence $\lambda(n)$ satisfies conditions $\mathcal{A}_2^\infty$, if $\mathbf{E}\lambda^2(n) \geq \psi(n) = \mathcal{L}(u_n) \to \infty$, where $\mathcal{L}(m)$ is a regularly varying sequence with exponent $3 - \beta \in (0.2)$ as $m \to \infty$, independent of $n$, $u_n \leq n/2$, and $u_n \to \infty$ is a regularly varying sequence as $n \to \infty$. Moreover, for some constant $C > 0$ independent of $n$ and any $b \in \mathbb{N}_{u_n} \backslash \mathbb{N}_{m_0}$ (for sufficiently large $n$), the following inequality holds:*

$$\mathbf{P}(b/2 \leq \lambda(n) \leq b) \geq Cb^{-2}\mathcal{L}(b). \tag{13}$$

The aim of this work is to generalize Theorems 5 and 6 from (Antipov et al., 2022) to wider classes of distributions of sequences of random variables $\lambda(n)$, $n \in \mathbb{N}$. Let us give the explicit form of distributions from (Antipov et al., 2022)

$$\mathbf{P}(\lambda(n) = k) = p_{n,k} = C_{\beta,u_n}k^{-\beta}, \; k \in \mathbb{N}_{u_n}, \tag{14}$$

where $u_n \to \infty$ as $n \to \infty$ and $C_{\beta,u_n} = \sum_{k=1}^{u_n} k^{-\beta}$. We consider only the case $\beta > 1$. Due to the convergence of the series $\sum_{k=1}^{\infty} k^{-\beta}$, the sequence $C_{\beta,u_n}$ is asymptotically constant. From the point of view of regularly varying functions, the probabilities $p_{n,k}$ from representation (14) are described in terms of regularly varying functions with exponent $-\beta$, in which the slowly varying factor is an asymptotically constant function of $u_n$. The second moment $\mathbf{E}\lambda^2(n) = C_{\beta,u_n} \sum_{k=1}^{u_n} k^{2-\beta}$, as shown in (Antipov et al., 2022), is of order $u_n^{3-\beta}$, i.e., it will be a regularly varying function with exponent $3 - \beta$ for $\beta < 3$. This motivates the choice of the parameter $3 - \beta$ in definition $\mathcal{A}_2^\infty$.

Condition $\mathcal{A}_2^c$ means that the second moment of the random variable $\lambda(n)$ is uniformly bounded and there are no other constraints on $p_{n,k}$, which is realized for $\beta > 3$ in condition (14).

Condition $\mathcal{A}_2^\infty$ means that the regularly varying at infinity function $\mathcal{L}(v)$ has the form $\mathcal{L}(v) = v^{3-\beta}\ell(v)$, $v \in \mathbb{R}^+$, where $\ell(v)$ varies slowly at infinity. Condition (13) can be written as

$$\mathbf{P}(b/2 \leq \lambda(n) \leq b) \geq Cb^{1-\beta}\ell(b).$$

In terms of assumptions (14), these conditions hold for $1 < \beta < 3$ and $\ell(v)$ asymptotically constant, but the explicit form of $p_{n,k}$, the probabilities of specific values for $\lambda(n)$, is not important, and only bounds for their sums over long intervals are used, which will be regularly varying at infinity functions with exponent $1 - \beta$. Obviously, $\mathcal{L}(u_n)$, as a composition of regularly varying sequences, will be a regularly varying sequence as $n \to \infty$ with a naturally computed exponent.

Condition $\mathcal{A}_2^\infty$ could also include some cases $\beta = -1, -3$, which yield explicit bounds for $\mathbf{E}\tau(n)$, but these proofs are cumbersome and require working with subtle structural theorems for slowly varying functions. Let us give a generalization of Theorem 5 from Antipov et al. (2022).

**Theorem 1** *The average number of iterations $\tau$ in algorithm $(1+(\lambda, \lambda))$ GA with the fitness function* ONEMAX *before reaching the optimum is*

$$\mathbf{E}\tau(n) \;=\; O(n \ln n), \; subject \; to \; \mathcal{A}_2^c; \tag{15}$$

$$\mathbf{E}\tau(n) \;=\; O(n) + O\left(\frac{n \ln n}{\psi(n)}\right), \; subject \; to \; \mathcal{A}_2^\infty, \tag{16}$$

*where $\psi(n)$ is from definition 4.*

Bound (16) for the number of iterations until the first hit of $x^*$ when passing to arbitrary regularly varying at infinity sequences $u_n$ and $\mathcal{L}(n) = n^{1-\beta}\ell(n)$ differs qualitatively from that obtained in Theorem 5 from (Antipov et al., 2022), where the second term is absent, which disappears when $\psi^{-1}(n) \ln n = \mathcal{L}^{-1}(u_n) \ln n = O(1)$. In the case of conditions (14), the last bound becomes $u_n^{\beta-3} \ln n = O(1)$, which implies condition $u_n \geq \ln^{1/(3-\beta)} n$ of Theorem 5 (Antipov et al., 2022).

## 4 UPPER BOUNDS FOR OPTIMIZATION TIME AND COMPUTATIONAL COST

Let $T^{op}$ be the number of computational operations until the first hit of $x^*$ in the RAM model with random access memory (Aho et al., 1974), where standard arithmetic operations have constant duration. The distributions of the random variables $T$ and $T^{op}$ depend on the dimension of the bit strings of individuals $n$. Therefore, for them, we further use the notations $T(n)$ and $T^{op}(n)$, respectively. Denote by $T_s(n)$ and $T_s^{op}(n)$ the number of fitness function evaluations and computational operations until the first hit of $x^*$, respectively, given that the process starts from an individual $x^{(0)} \in Z_s$.

Denote the number of fitness function evaluations and the number of computational operations in one iteration, starting from an individual $x \in Z_s$, ending with the preservation, possibly of a different individual $x$ from $Z_s$ for the next operation, for fixed $\lambda_{i,s}(n)$, by $\mu_{\lambda(n)}(i, s)$ and $\nu_{\lambda(n)}(i, s, n)$. Here the random variables $\lambda_{i,s}(n)$ are independent in $i$ and $s$ and have the same distribution as $\lambda(n)$, and the index $i$ denotes the ordinal number of a start from an individual $x$ (possibly different from the original one, but with the same fitness function value) with an unsuccessful outcome – without transition to a higher level. The averages over $\lambda_{i,s}(n)$ for $\mu_{\lambda(n)}(i, s)$ and $\nu_{\lambda(n)}(i, s, n)$ are denoted by $\mu_n(i, s) = \mathbf{E}_{\lambda(n)}\mu_{\lambda(n)}(i, s)$ and $\nu_n(i, s) = \mathbf{E}_{\lambda(n)}\nu_{\lambda(n)}(i, s, n)$, respectively. Due to the identical distribution of the r.v.s $\lambda_{i,s}(n)$ in $i$, the latter averages coincide for different $i$ with fixed $s$ and $n$.

For $\mu_{\lambda(n)}(i, s)$ in the first step of the loop (mutation of the individual $x$), $\lambda(n)$ evaluations of the fitness function $\mathbf{E}\lambda(n) \leq \mu_n(i, s)$ are performed, and in the second step, corresponding to crossover, there are no more than $\lambda(n)$ such evaluations. Therefore, the constraints hold

$$\mu_n(i, s) \leq 2\mathbf{E}\lambda(n). \tag{17}$$

Estimates for $\nu_{\lambda(n)}(i, s)$ are more complicated. This value depends on the implementation of the computational algorithm. Suppose that there exists a function $\phi(\lambda(n), n)$ such that

$$\nu_{\lambda(n)}(i, s, n) \leq \phi(\lambda_{i,s}(n), n).$$

Denoting $\phi_n = \mathbf{E}_{\lambda(n)}\phi(\lambda_{i,s}(n), n)$, we obtain the inequalities

$$\nu_n(i, s) \leq \phi_n. \tag{18}$$

Let $\mu_{\lambda(n)}(s)$ and $\nu_{\lambda(n)}(s, n)$ denote the random numbers of fitness function evaluations and computational operations in one iteration starting from any individual $x$ in $Z_s$, ending with a transition to another individual with a higher fitness function value. Their averages over $\lambda(n)$ are denoted by $\mu_n(s) = \mathbf{E}_{\lambda(n)}\mu_{\lambda(n)}(s)$ and $\nu_n(s) = \mathbf{E}_{\lambda(n)}\nu_{\lambda(n)}(s, n)$, respectively. For these averages, inequalities analogous to (17) and (18) hold

$$\mu_n(s) \leq 2\mathbf{E}\lambda(n), \quad \nu_n(s) \leq \phi_n. \tag{19}$$

Let $\zeta_n(s)$ and $\eta_n(s)$ be the number of objective function evaluations and the number of computational operations during the time that successive individuals $x$ stay in $A_s$, ending with a transition to an individual with a higher fitness function value. More specifically, the representations hold

$$\zeta_n(s) = \sum_{i=1}^{\ell_n(s)-1} \mu_n(i, s) + \mu_n(s), \quad \eta_n(s) = \sum_{i=1}^{\ell_n(s)-1} \nu_n(i, s) + \nu_n(s), \tag{20}$$

where the sum is zero if the upper index is less than the lower one.

Relations (17), (18), (19) and (20) and Kolmogorov-Prokhorov theorem (Borovkov, 1976, Ch. 4, §4) for non-negative random variables allow us to write the inequalities

$$\mathbf{E}\ell_n(s)\mathbf{E}\lambda(n) \leq 2\mathbf{E}\ell_n(s)\mathbf{E}\lambda(n), \quad \mathbf{E}\eta_n(s) \leq \mathbf{E}\ell_n(s)\phi_n. \tag{21}$$

The next theorem is a generalization of Theorem 6 from (Antipov et al., 2022).

**Theorem 2** *The average number of fitness function evaluations $T(n)$ and the number of operations $T^{op}(n)$ in algorithm $\mathcal{A}$ with fitness function ONEMAX are bounded from above by the quantities*

$$\mathbf{E}T(n) \quad = \quad O(n \ln n), \text{ under condition } \mathcal{A}_2^c;$$

$$\mathbf{E}T(n) = O\left(n + \frac{n\ln n}{\psi(n)}\right)\mathbf{E}\lambda(n), \textit{ under condition } \mathcal{A}_2^{\infty};$$ (22)

$$\mathbf{E}T^{op}(n) = O(n\ln n)\phi_n, \textit{ under condition } \mathcal{A}_2^{c};$$

$$\mathbf{E}T^{op}(n) = O\left(n + \frac{n\ln n}{\psi(n)}\right)\phi_n, \textit{ under condition } \mathcal{A}_2^{\infty}.$$

The statement of Theorem 6 (Antipov et al., 2022) corresponds to the special case of bounds (22) under the conditions

$$\mathbf{E}\lambda(n) \le C < \infty, \ \forall n \in \mathbb{N},$$

(or $\beta > 2$), $\psi(n) = u_n^{3-\beta}$ and $u_n \ge \ln^{1/(3-\beta)} n$, which is a special case of the condition $u_n^{\beta-3}\ln n = O(1)$.

**Corollary 1** *If we choose the set $\mathbb{N}^{(2)} = \{2^\ell, \ell = 0, 1, 2, \cdots\}$ as the support of the distribution of $\lambda(n)$ and set $p_{n,k} := C^{(*)}(\beta, u_n)k^{1-\beta}$ for $k \in \mathbb{N}_{u_n}^{(2)}$ and $u_n \in \mathbb{N}^{(2)}$, then $\mathbf{E}T(n) = O(n)$ in the case of the fitness* ONEMAX.

In the next section, an experimental comparison of CPU time until the first reaching of the optimum of the ONEMAX function is carried out when generating the r.v. $\lambda(n)$ as indicated in Corollary 1 or according to the power law with support $\{0, 1, 2, \ldots, n\}$ from (Antipov et al., 2022).

## 5 COMPUTATIONAL EXPERIMENT

In the computational experiment, algorithm $\mathcal{A}$ was considered, where the r.v. $\lambda(n)$ was generated as indicated in Corollary 1 (hereinafter, algorithm A) and the algorithm $(1 + (\lambda, \lambda))$ GA, where $\lambda(n)$ is chosen according to the power law with support $\{0, 1, 2, \ldots, n\}$ from (Antipov et al., 2022) (hereinafter, algorithm B). The optimization criterion was the function ONEMAX, and the algorithm had an adjustable parameter $\beta = 2.75$ and an upper bound on the number of offspring $u_n = n$.

The software code in Scala proposed by the authors of (Antipov et al., 2022) was used as a basis. Changes concerned only the procedure for generating $\lambda(n)$: in the case of algorithm A, it can be implemented with complexity $O(\log\log(u_n))$, while in the original implementation from (Antipov et al., 2022), choosing $\lambda$ requires $O(\log(u_n))$ operations. In order to compare the computational costs in CPU time until the first reaching the optimum, experiments were carried out for $n = 2^{15}, 2^{16}, 2^{17}, 2^{18}$ and $2^{19}$. Computations were performed on a server with an AMD EPYC 7502 processor using 5 independent parallel threads. For each instance, both algorithms were run $10^5$ times. The arithmetic mean $\hat{T}_A, \hat{T}_B$ and the estimates of standard deviation $\hat{\sigma}_A, \hat{\sigma}_B$ for the measured computation time (milliseconds) are given in Table 1. As can be seen from the table, algorithm A has an advantage in average computation time and smaller standard deviation.

| $n$ | $2^{15}$ | $2^{16}$ | $2^{17}$ | $2^{18}$ | $2^{19}$ |
|---|---|---|---|---|---|
| $\hat{T}_A$ | 93.91 | 201.65 | 476.82 | 1141.03 | 2790.51 |
| $\hat{\sigma}_A$ | 225.58 | 593.88 | 2005.57 | 6629.19 | 22902.68 |
| $\hat{T}_B$ | 117.11 | 266.83 | 608.05 | 1492.12 | 3783.67 |
| $\hat{\sigma}_B$ | 420.09 | 1495.47 | 5007.79 | 18652.82 | 64356.55 |

Table 1: Average CPU time (ms) until first hitting the optimum and the estimate of its standard deviation for algorithms A and B depending on the problem size.

## 6 CONCLUSION

In this paper, we investigated the question of the average time to obtain the optimal solution of a simple model problem of maximizing ONEMAX using a known version of the genetic algorithm $(1 + (\lambda, \lambda))$ GA. The main result of this work is that the upper bound obtained in the article by Antipov, Buzdalov and Doerr (Antipov et al., 2022) for $(1 + (\lambda, \lambda))$ GA with the fast mutation operator remains valid in a more general case, when the distributions for the size of the intermediate

population $\lambda$ and the mutation parameter $p$ are chosen from a wider class of distributions. The conducted computational experiment showed promising results, suggesting that the proposed method of random selection of the population size $\lambda(n)$ may be useful in practice.

The function ONEMAX considered here has only one local optimum, which is also global. As follows from the theoretical results obtained in (Doerr et al., 2017) for $(1 + (\lambda, \lambda))$ GA on the model function JUMP with many local optima, the use of fast mutation on this function removes the problem of exact tuning of the population size and mutation parameter. In this regard, in further research it makes sense to consider the possibilities of relaxing the requirements for the distribution of these parameters in the fast mutation operator when optimizing the JUMP function and other multimodal functions, in particular, when solving NP-hard pseudo-Boolean optimization problems.

## ACKNOWLEDGMENTS

This research was funded by the Mathematical Center in Akademgorodok under the agreement № 075-15-2025-349 with the Ministry of Science and Higher Education of the Russian Federation.

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

APPENDIX

This appendix contains the proofs omitted from the main text of the paper.

**Lemma 2.** *In the case of the fitness function* ONEMAX*, the average number of iterations of algorithm* $\mathcal{A}$*, starting from a solution* $x \in Z_s$*, until improvement of the objective function, under condition (7) satisfies the inequality*

$$\mathbf{E}\ell_n(s) \leq C_4^{-1} n/s, \ s \in \mathbb{N}_n, \tag{23}$$

*where* $C_4$ *is defined in expression (8).*

**Proof.**     Estimate   (23)   follows   from   relations   (6)   and   (8).        Q.E.D.

**Theorem 1.** *The average number of iterations* $\tau$ *in algorithm* $(1 + (\lambda, \lambda))$ *GA with the fitness function* ONEMAX *before reaching the optimum is*

$$\mathbf{E}\tau(n) \ = \ O(n \ln n), \ \text{subject to } \mathcal{A}_2^c; \tag{24}$$

$$\mathbf{E}\tau(n) \ = \ O(n) + O\left(\frac{n \ln n}{\psi(n)}\right), \ \text{subject to } \mathcal{A}_2^\infty, \tag{25}$$

*where* $\psi(n)$ *is from definition 4.*

**Proof.** Note that under condition $\mathcal{A}_2^c$, the first moment $\mathbf{E}\lambda(n)$ is also uniformly bounded in $n$. Consequently, by Lemma 2, under conditions $\mathcal{A}_2^c$, the inequality holds

$$\mathbf{E}\ell_n(s) \leq C_4^{-1}\frac{n}{s}. \tag{26}$$

Note that the sum $\sum_{s=1}^u s^{-1}$ of a regularly varying step function as $u \to \infty$ will be asymptotically equivalent to the integral $\int_1^u x^{-1}dx$, i.e., as $u \to \infty$

$$\sum_{s=1}^u s^{-1} \sim \int_1^u x^{-1}dx = \ln u. \tag{27}$$

Estimate (24) follows from relations (11), (26) and (27).

Now we prove inequality (25).

Consider first the case $u_n^2 s/n < 1$. Under condition $\mathcal{A}_2^\infty$, bound (4) leads to the inequality

$$p_n(s) = \mathbf{E}_{\lambda(n)} p_{\lambda(n)}(s) \geq C^* \mathcal{L}(u_n)s/n = C^*\psi(n)s/n. \tag{28}$$

Therefore, for $u_n^2 s/n < 1$ and under condition $\mathcal{A}_2^\infty$, from relations (6) and (28) we obtain

$$\mathbf{E}\ell_n(s) \leq C^* \frac{n}{\psi(n)s}, \tag{29}$$

where formally instead of $C^*$ should be $1/C^*$ with the constant from bound (28), but in accordance with our assumptions we use the same notation $C^*$ for it.

This bound for $s > n\epsilon$, where $\epsilon \in (0, 1)$ is an arbitrary fixed number, and $u_n^2 \to \infty$, is not applicable, but if $s > n\epsilon$, then the sum containing such terms $\mathbf{E}\ell_n(s)$ has already been boundd above by the quantity $Cn$ in relations (11) and cannot be improved in order.

Now consider the case $u_n^2 s/n \geq 1$. In this case, under conditions $\mathcal{A}_2^\infty$, from inequality (13) for $m_0 < b \leq u_n$ follows the bound

$$\mathbf{E}_{\lambda(n)} \lambda^2(n) \geq \mathbf{E}_{\lambda(n)}\left\{\lambda^2(n); \lambda(n) \in [b/2, b]\right\} \geq C\mathcal{L}(b). \tag{30}$$

In the case under consideration, for sufficiently large $n$, the inequalities $m_0 < \sqrt{n/s} \leq u_n$ hold. Therefore, using inequality (30) with $b = \sqrt{n/s}$, $s \in \mathbb{N}_{\epsilon n}$, we bound the average $p_{\lambda(n)}(s)$ on the set $\lambda^2(n) < n/s$

$$\mathbf{E}_{\lambda(n)} p_{\lambda(n)}(s) \geq C\mathcal{L}(b)s/n.$$

From here and from relation (6) we obtain

$$\mathbf{E}\ell_n(s) \leq C^* \frac{n}{\mathcal{L}(\sqrt{n/s})s}. \tag{31}$$

Inequalities (29) and (31), obtained in the two cases considered above, imply the bound

$$\sum_{s=1}^{n\epsilon} \mathbf{E}\ell_n(s) \quad \leq \quad \frac{C^*n}{\psi(n)} \sum_{s=1}^{n/u_n^2-1} s^{-1} + C^* \sum_{s=n/u_n^2}^{n\epsilon} \frac{n}{\mathcal{L}(\sqrt{n/s})s} \tag{32}$$

$$\leq \quad C^* \frac{n}{\psi(n)} \ln(n/u_n^2) + C_1^* n. \tag{33}$$

We explain inequality (33). The first sum from the right-hand side of expression (32) is bounded here using relation (27). The terms of the second sum from (32) can be represented via the function $\mathcal{L}_0(x) := x\mathcal{L}^{-1}(\sqrt{x})$, where $x = n/s$, which will be regularly varying at infinity with exponent $\beta_0 := 1 - 0.5(3 - \beta) = 0.5(\beta - 1)$. Under the change of variables $y = x^{-1}$, the function $\mathcal{L}_0(y^{-1})$ will be regularly varying at zero with exponent $-1 < -\beta_0 < 0$. From the function $\mathcal{L}_0(y^{-1})$, we determine an equivalent regularly varying at zero function $\tilde{\mathcal{L}}_0(y^{-1})$ with values $\mathcal{L}_0(s/n)$ on the set $y^{-1} \in [s/n, (s+1)/n)$. The sum of a regularly varying sequence $\mathcal{L}_0(s/n)$ coincides with the integral of $\tilde{\mathcal{L}}_0(y^{-1})$, which is constant on half-intervals of length $n^{-1}$ and will be regularly varying at zero with exponent $-1 < -\beta_0 < 0$. After the change of variables $u = yn^{-1}$, the last integral becomes

$$n \int_{u_n^{-2}}^{\epsilon} \tilde{\mathcal{L}}_0(u)du \leq n \int_0^{\epsilon} \tilde{\mathcal{L}}_0(u)du,$$

which converges at zero for $-1 < -\beta_0 < 0$ by the lemma from (Feller, 1971, Ch. VIII, §9). The uniform boundedness of the second sum is proved, which completes the proof of bound (33).

If $\psi(n)$ varies regularly with a positive exponent, then the first term in (33) will be of order $o(n)$, and the integral will converge. As a result, under conditions $\mathcal{A}_2^{\infty}$ for functions $\psi(n)$ that are not slowly varying, we have

$$\sum_{s=1}^{n\epsilon} \mathbf{E}\ell_n(s) \leq C^* n. \tag{34}$$

For slowly varying unboundedly growing functions $u_n$, by property $2°$ from (Seneta, 1976, Ch. 1, Sec. 1.5), the relation $\ln u_n = o(\ln n)$ holds, which together with bounds (11) and (33) proves relation (25). Q.E.D.

**Theorem 2.** *The average number of fitness function evaluations $T(n)$ and the number of operations $T^{op}(n)$ in algorithm $\mathcal{A}$ with fitness function* ONEMAX *are bounded from above by the quantities*

$$\mathbf{E}T(n) \quad = \quad O(n\ln n), \text{ under condition } \mathcal{A}_2^c;$$

$$\mathbf{E}T(n) \quad = \quad O\left(n + \frac{n\ln n}{\psi(n)}\right) \mathbf{E}\lambda(n), \text{ under condition } \mathcal{A}_2^{\infty}; \tag{35}$$

$$\mathbf{E}T^{op}(n) \quad = \quad O(n\ln n)\phi_n, \text{ under condition } \mathcal{A}_2^c;$$

$$\mathbf{E}T^{op}(n) \quad = \quad O\left(n + \frac{n\ln n}{\psi(n)}\right)\phi_n, \text{ under condition } \mathcal{A}_2^{\infty}.$$

**Proof.** Applying the law of total probability by analogy with (11) and using inequalities (21), we have

$$\mathbf{E}T(n) = 2^{-n} \sum_{s=0}^{n-1} C_n^s \mathbf{E}T_s(n) \leq C^* \mathbf{E}\tau(n)\mathbf{E}\lambda(n), \tag{36}$$

$$\mathbf{E}T^{op}(n) = 2^{-n} \sum_{s=0}^{n-1} C_n^s \mathbf{E}T_s^{op}(n) \leq C^* \mathbf{E}\tau(n)\phi_n. \tag{37}$$

Theorem 1 and bounds (36) and (37) imply the statement of Theorem 2. Q.E.D.

