# OpenReview forum: "Generalized Heavy-Tailed Mutation for Evolutionary Algorithms"
_mathai.club/MathAI/2026/Conference — 2026 Oral_

### Official Review · Reviewer_tSau · 2026-03-11
**This submission presents an incremental and insufficiently validated generalization of prior heavy-tailed mutation results for the \((1+(\lambda,\lambda))\) GA. While the stated goal is to replace a specific truncated power-law assumption with broader regularly varying conditions, the paper does not provide proofs, does not demonstrate convincing empirical significance, and does not make a compelling case that the generalization yields meaningful new algorithmic insight beyond the existing literature.**

**Rating:** 2
**Confidence:** 4

**Review:**

# Review

## Summary

This paper studies the \((1+(\lambda,\lambda))\) genetic algorithm on **ONEMAX** under a generalized heavy-tailed mutation scheme. The main claim is that prior upper bounds established for power-law-distributed \(\lambda\) can be extended to a broader class of distributions characterized by regularly varying constraints. The paper also proposes a new sampling scheme for \(\lambda\) and reports CPU-time experiments.

In principle, the mathematical direction is reasonable: replacing the exact assumption
\[
\Pr(\lambda = k) \propto k^{-\beta}
\]
with interval-mass lower bounds of the form
\[
\Pr(b/2 \le \lambda \le b) \ge C b^{-2} L(b),
\]
where \(L\) is regularly varying, could be a meaningful abstraction. However, in its current form, the paper is **not ready for publication**.

---

## Overall evaluation

### Quality: **Low**

The paper’s central problem is that the main theoretical claims are **not verifiable from the manuscript**. The authors explicitly state that the proofs are omitted. For a paper whose entire contribution is a theoretical generalization, this is fatal. Without proofs, there is no way to assess whether the regularly varying framework has been applied correctly, whether the constants and asymptotic transitions are justified, or whether hidden assumptions have been introduced.

The empirical section also does not rescue the paper. It evaluates only **ONEMAX**, only compares CPU time, and appears to conflate algorithmic improvement with the engineering advantage of a cheaper sampling routine. That is not convincing evidence of a meaningful optimization advance.

### Clarity: **Poor**

The exposition is difficult to follow and contains multiple notation and presentation issues:

- assumptions and symbols are introduced in a compressed and sometimes inconsistent way;
- some definitions are malformed or appear to contain typographical errors;
- the logical bridge from the assumptions to Theorems 1 and 2 is too thin to audit;
- the experimental section does not clearly isolate what is actually being tested.

A theory paper of this kind must be unusually clean and explicit. This manuscript is not.

### Originality: **Moderate at best**

The contribution is best described as an **incremental abstraction** of Antipov et al. (2022), not a fundamentally new algorithmic or analytical idea. The paper does not introduce a new problem, a new analysis paradigm of broad independent value, or a new performance phenomenon. It mostly argues that the exact power-law form can be weakened to a regularly varying condition. That may be mathematically legitimate, but the novelty is limited and the presentation does not elevate it into a substantial conceptual advance.

### Significance: **Low**

Even if the theorems are correct, the significance is limited for several reasons:

1. the analysis remains confined to **ONEMAX**, the easiest benchmark in this area;
2. the practical evidence is minimal;
3. the claimed computational advantage is likely due to **faster generation of \(\lambda\)** rather than a better search dynamic;
4. the paper does not show that the generalized framework yields better understanding or better performance on genuinely difficult multimodal landscapes.

The manuscript repeatedly gestures toward broader relevance (e.g., jump functions, NP-hard pseudo-Boolean optimization), but none of that is actually demonstrated.

---

## Main strengths

### Pros

- The paper addresses a natural theoretical question: whether heavy-tailed mutation results depend on the exact truncated power-law form or only on more general tail regularity.
- The regularly varying viewpoint is mathematically plausible and, in principle, could unify several mutation distributions under one umbrella.
- The corollary using dyadic support is potentially interesting from an implementation standpoint, because it may reduce the cost of sampling \(\lambda\).
- The paper is positioned in a recognizable line of work and cites relevant prior results.

---

## Main weaknesses

### Cons

1. **No proofs for the main theorems.**
   This is the single most serious flaw. A theorem-driven paper without proofs is not evaluable. In a result of the form
   \[
   \mathbb{E}[\tau(n)] = O(n) + O\!\left(\frac{n \log n}{\psi(n)}\right),
   \]
   the details matter completely. Omitting them prevents any serious assessment of correctness.

2. **The novelty is incremental and heavily derivative.**
   The work is essentially a generalization of known bounds from a specific heavy-tailed law to a broader regularly varying family. That is a modest extension, not a major advance.

3. **The paper is too dependent on ONEMAX.**
   ONEMAX is a standard sanity-check benchmark, not strong evidence of broader impact. A paper motivating generalized heavy-tailed mutation should show why the generalization matters beyond the easiest unimodal case.

4. **The experiments are weak and partially misaligned with the claim.**
   The reported benefit is CPU time, but the modified method also changes the complexity of generating \(\lambda\). Thus the experiment does not cleanly show a search advantage. It mostly suggests that one implementation is cheaper than another.

5. **No evaluation on the quantity that matters most theoretically.**
   If the main theoretical story is about optimization time / function evaluations, then the experiments should focus primarily on:
   - number of fitness evaluations,
   - hitting-time distributions,
   - scaling with \(n\),
   - comparison under equalized sampling overhead.

6. **Large variance, weak statistical reporting.**
   The reported standard deviations are extremely large relative to the means. There are no confidence intervals, no significance tests, no distribution plots, and no discussion of whether the observed gaps are robust.

7. **Several notation and editorial problems undermine trust.**
   Examples include malformed interval notation, overloaded symbols, awkward phrasing, and apparent typographical issues in the assumptions. In a technical asymptotic analysis, these are not cosmetic problems; they directly affect interpretability.

8. **Important implementation-dependent terms are left vague.**
   The analysis introduces quantities such as \(\phi_n\) for operation counts, but these are not instantiated in a way that yields a clear, practically meaningful theorem.

9. **The paper overstates practical promise.**
   The phrase “promising results” is too strong given that the evidence is limited to ONEMAX and modest constant-factor CPU-time differences.

10. **The broader motivation is not cashed out.**
    The conclusion suggests relevance to multimodal functions and NP-hard problems, but the paper provides no theorem or experiment supporting those directions.

---

## Detailed concerns

### 1. Theoretical contribution is impossible to verify

The paper’s value rests almost entirely on Theorems 1 and 2. But a theorem-only paper without proofs is not acceptable unless the results are straightforward corollaries of well-known arguments, which does not seem to be the case here. The move from explicit power-law probabilities to regularly varying interval constraints is exactly the place where subtle asymptotic errors can occur.

In particular, the manuscript asks the reviewer to trust several nontrivial transitions of the form:
\[
p_n(s) = \mathbb{E}_{\lambda(n)}[p_{\lambda(n)}(s)],
\qquad
\mathbb{E}[\ell_n(s)] = p_n(s)^{-1},
\]
and then derive global bounds on hitting times under generalized tail assumptions. These are the core arguments, yet none are shown.

### 2. The regular variation framework is under-explained

The paper presents definitions of regular variation, but the role of these assumptions in the algorithmic analysis is not developed clearly enough. The reader is left with the impression that regular variation is being used mainly as a broad asymptotic label rather than as a sharply exploited analytical tool.

A stronger paper would explain:

- exactly which step of the Antipov et al. proof uses only tail mass over intervals;
- why regular variation is the right abstraction rather than just one possible one;
- whether the result is close to maximal, or whether even weaker assumptions would suffice.

### 3. Experimental evidence does not support the main narrative

The proposed algorithm A appears faster in CPU time than algorithm B, but the paper itself notes that the sampling procedure for \(\lambda(n)\) can be implemented in
\[
O(\log \log u_n)
\]
instead of
\[
O(\log u_n).
\]
This means the observed CPU gain may simply reflect cheaper random sampling, not a stronger optimization process. That is an implementation result, not evidence that the generalized mutation law improves evolutionary search.

To support the paper’s claims, the authors should at minimum compare:

- **fitness evaluations to optimum**, not just wall-clock time;
- **identical sampling overhead conditions**;
- **multiple benchmark landscapes**, especially ones where heavy-tailed effects are actually known to matter.

### 4. The significance is overstated

The manuscript suggests that the result is practically meaningful because \(O(n)\) is asymptotically better than the best static-mutation choice. But this is already the established motivation in the prior literature. The present paper does not demonstrate a qualitatively new phenomenon; it only widens the admissible family of \(\lambda\)-distributions.

That may be mathematically fine, but then the paper should sell itself honestly as a **technical extension**, not as a substantial new development.

---

## What would be needed for acceptance

This paper would need major revision before it could be considered publishable:

1. **Include full proofs** of the main theorems, or at least a complete appendix.
2. **Clean up the notation and assumptions** carefully.
3. **Clarify the exact novelty** relative to Antipov et al. (2022).
4. **Expand experiments** beyond ONEMAX.
5. **Separate search quality from sampling overhead** in the empirical comparison.
6. **Report evaluation counts and proper statistical uncertainty**, not only mean CPU time.
7. **Provide a clearer conceptual message**: what does regular variation teach us that the power-law analysis did not?

---

## Final recommendation

**Reject.**

The central idea is not unreasonable, but the current paper is not competitive as a scientific submission. The theory is not reviewable because the proofs are absent, the empirical support is too weak to compensate, the novelty is incremental, and the manuscript does not yet meet the standards of clarity and rigor required for a theorem-driven paper.

---

> ### Author Rebuttal · Authors · 2026-03-12
>
> We agree that the proofs need to be in the paper. Here you can download this paper with the proofs provided in the appendix:
> https://disk.yandex.ru/i/O3Y2YYIPpfdWig
> If it gets accepted, the appendix will be available as supplementary materials, or included into the paper.

---

### Official Review · Reviewer_9BjS · 2026-03-11
**Promising theoretical extension, but requires full proofs and broader empirical validation**

**Rating:** 5
**Confidence:** 4

**Review:**

The paper presents a theoretical generalization of the heavy-tailed mutation operator  for the $(1+(\lambda,\lambda))$ genetic algorithm, specifically analyzing its expected optimization time on the OneMax problem. The authors replace the standard power-law assumption with a broader regularly varying constraint on the mutation rate distribution. This is a mathematically sound direction, but the current paper has notable limitations that hinder its overall impact.On the positive side, the theoretical framing is good. By leveraging regularly varying functions, the authors effectively extend the sufficient conditions while preserving the linear $O(n)$ expected optimization time under specific constraints. Furthermore, the computational experiment highlights a genuine practical advantage: the proposed generalized formulation allows for a more efficient sampling of the population size $\lambda$, reducing the operational complexity to $O(\log \log u_n)$ compared to the previous $O(\log u_n)$. This translates to empirically faster CPU times when evaluated on the OneMax benchmark for large problem sizes up to $n=2^{19}$

But, the authors explicitly state that the proofs for the main theorems are omitted due to page limits. For a mathematically-driven paper, omitting the core proofs makes it impossible to rigorously verify the theoretical claims during the review process. Additionally, the empirical validation is extremely limited. The experiments and theoretical bounds are exclusively focused on the OneMax function, which is a unimodal and highly simplified benchmark.

To improve the paper, the authors really need to find a way to include the full proofs. A straightforward solution to free up space would be to condense the standard background material on regularly varying functions, or simply defer the proofs to a supplementary appendix. Without these details, the core theoretical claims cannot be properly verified.

Furthermore, while the authors correctly identify the JUMP function as a relevant target in their conclusion, they really should include it in the current empirical evaluation rather than leaving it for future work. Relying solely on OneMax is a good theoretical starting point, but it isn't entirely convincing for a modern EA paper. The study would be significantly more impactful if the generalized mutation operator were tested against a broader suite of challenging environments—such as deceptive Trap functions , rugged NK-landscapes, or classic NP-hard pseudo-Boolean optimization tasks like MAX-3SAT and Minimum Vertex Cover.

---

### Official Review · Reviewer_2H2z · 2026-03-11
**Promising idea, but missing proofs and limited experiments**

**Rating:** 5
**Confidence:** 2

**Review:**

\textbf{Strengths.} The paper addresses a meaningful theoretical question and the proposed direction appears mathematically natural. The motivation is clear, and the paper suggests that the generalized construction may also have practical implementation benefits due to cheaper sampling.

\textbf{Weaknesses.} My main concern is that the central theoretical claims are not fully verifiable from the submitted version, because the proofs of the main theorems are omitted. For a mathematically oriented paper, this is a serious issue. In addition, the empirical validation is rather narrow and focuses essentially only on \textsc{OneMax}. While \textsc{OneMax} is a standard baseline benchmark for evolutionary algorithms, results only on this function are not fully convincing for evaluating the broader usefulness of the proposed mutation scheme.

I also note that the practical evidence seems to emphasize CPU-time improvements, which may partly reflect cheaper sampling rather than a stronger optimization effect in a broader sense. This does not invalidate the work, but it limits the strength of the empirical claims in the current form.

\textbf{Recommendation.} Overall, I find the topic interesting, but in its present form the paper is slightly below the acceptance threshold. The main reasons are the lack of full proofs in the submitted manuscript and the limited empirical validation.

---

### Decision · Program_Chairs · 2026-03-14

**Decision:**

Accept (Oral)

**Comment:**

Dear Author(s),

On behalf of the Program Committee of the International Conference on Mathematics of Artificial Intelligence (MathAI 2026), we are pleased to inform you that your paper has been accepted for an oral presentation at MathAI 2026.

Your paper was evaluated through a rigorous two-stage review process involving both automated screening and expert review by members of the Program Committee. The reviewers recognized the quality and contribution of your work.

Presentation details:

- Format: Oral presentation (15–20 minutes + 5 minutes Q&A)
- Mode: You may present either in person (offline) at the conference venue in Sirius, Russia, or remotely via Zoom. Please indicate your preferred mode when confirming your participation.
- Conference dates: Marh 30 - April 3, 2026
- Website: https://mathai.club

Next steps:

1. Please confirm your participation and presentation mode by replying to this email mathai.club@yandex.ru no later than March 15, 2026 18:00 Moscow time.
2. If you plan to attend in person, the organizing committee will provide accommodation details separately.
3. Please prepare your final camera-ready manuscript according to the formatting guidelines available at https://mathai.club and upload it to OpenReview by March 15, 2026 18:00 Moscow time.

Should you have any questions regarding the program, logistics, or your presentation slot, please do not hesitate to contact us.

We look forward to your contribution to MathAI 2026.

With kind regards,

MathAI 2026 Program Committee
International Conference on Mathematics of Artificial Intelligence
https://mathai.club
OpenReview: https://openreview.net/group?id=mathai.club/MathAI/2026/Conference
Telegram: https://t.me/MathAI_club
Email: mathai.club@yandex.ru